# Preparation of Nanocrystals for Insoluble Drugs by Top-Down Nanotechnology with Improved Solubility and Bioavailability

**DOI:** 10.3390/molecules25051080

**Published:** 2020-02-28

**Authors:** Xun Zhang, Zhiguo Li, Jing Gao, Zengming Wang, Xiang Gao, Nan Liu, Meng Li, Hui Zhang, Aiping Zheng

**Affiliations:** Department of Pharmaceutics, Institute of Pharmacology and Toxicology, Academy of Military Medical Sciences, Beijing, China, 27th Taiping Road, Haidian District, Beijing 100850, Chinaziguolee@163.com (Z.L.); gjsmmu@126.com (J.G.); wangzm.1986@163.com (Z.W.); gaoxiang609@163.com (X.G.); wowlinan@sohu.com (N.L.)

**Keywords:** midazolam, nanocrystals, pharmacokinetics, anticonvulsant, pharmacodynamics

## Abstract

Midazolam is a rapidly effective benzodiazepine drug that is widely used as a sedative worldwide. Due to its poor solubility in a neutral aqueous solution, the clinical use of midazolam is significantly limited. As one of the most promising formulations for poorly water-soluble drugs, nanocrystals have drawn worldwide attention. We prepared a stable nanosuspension system that causes little muscle irritation. The particle size of the midazolam nanocrystals (MDZ/NCs) was 286.6 ± 2.19 nm, and the crystalline state of midazolam did not change in the size reduction process. The dissolution velocity of midazolam was accelerated by the nanocrystals. The pharmacokinetics study showed that the AUC0–t of the MDZ/NCs was 2.72-fold (*p* < 0.05) higher than that of the midazolam solution (MDZ/S), demonstrating that the bioavailability of the MDZ/NC injection was greater than that of MDZ/S. When midazolam was given immediately after the onset of convulsions, the ED_50_ for MDZ/NCs was significantly more potent than that for MDZ/S and DZP/S. The MDZ/NCs significantly reduced the malondialdehyde content in the hippocampus of the seizures model rats and significantly increased the glutathione and superoxide dismutase levels. These results suggest that nanocrystals significantly influenced the dissolution behavior, pharmacokinetic properties, anticonvulsant effects, and neuroprotective effects of midazolam and ultimately enhanced their efficacy in vitro and in vivo.

## 1. Introduction

Epilepsy is the one of most common diseases and affects over 65 million people around the world, creating a significant burden due to epilepsy-related disabilities, stigmas, comorbidities, and costs [1,2]. Epilepsy affects the entire age range, from newborn infants to the elderly. It has many causes and manifestations, some of which are recognizable syndromes and some of which are still not clearly classified [3]. Currently, the main antiepileptic drugs used in the clinic are diazepam, gabapentin, clonazepam, and phenytoin sodium. Although these drugs have a clear therapeutic effect on epilepsy, their use is limited due to drug resistance [4]. One study estimates that up to 22.5% of epileptics have drug resistance, which signifies they are at increased risk of premature death, injury, psychosocial dysfunction, and poor quality of life [5,6,7].

Midazolam (MDZ) is a new type of ultra-short-effect benzodiazepine that has a low absorption rate, few accumulation phase pairs, and few toxic side effects [8]. It has been recently reported that the continuous intramolecular application of midazolam can significantly inhibit the status of convulsion and epilepsy [9]. In particular, when convulsions cannot be inhibited by diazepam and phenytoin sodium, midazolam is still effective, peculiarly in the early stages of convulsions.

Nanocrystals, which are a type of submicron colloidal dispersion system with a mean particle size between 10 and 1000 nm, have recently been applied to overcome the formulation issues of poorly soluble drugs and have attracted wide attention [10,11,12]. Nanometer-scale particles have a high relative surface area, and nanocrystals can greatly increase the dissolution velocity and saturation solubility of insoluble drugs, consequently improving their bioavailability [13]. Carrier-free drug delivery systems contain only pure drug crystals and a minimum amount of stabilizers [14,15,16]. Currently, there are several commercially available brands of drug nanocrystals on the market involving oral and injectable formulations. Different nanocrystals have different predilections toward delivering insoluble drugs. Thus, nanocrystal formulation was selected as a way to apply midazolam for possible clinical use.

In this study, we prepared midazolam nanocrystals via wet milling with different stabilizers to obtain qualified, stable, and low-muscle-irritation nanosuspensions. The physicochemical characterizations of midazolam nanoparticles (MDZ/NCs) were analyzed using scanning electron microscopy (SEM), X-ray diffraction (XRD), and differential scanning calorimetry (DSC). After determining the necessary prescription, we evaluated the midazolam nanoparticles (MDZ/NCs) and midazolam solution (MDZ/S) via in vivo and in vitro pharmacokinetic and pharmacodynamic experiments.

## 2. Results

### 2.1. Particle Size Distribution (PSD) of the Freshly Milled Product

The purpose of this study is to identify the relevant material property variables (the MDZ and the stabilizer). Nine stabilizers were selected, with a range of physical properties. The MDZ-related properties that were studied include molecular weight, melting point, solubility, logP, pKa, enthalpy, and morphology (Found in Pubchem database), as shown in Table 1.

Surface tension measurements of the stabilizer solutions were determined using the Wilhelmy plate method on a DCAT21 instrument (Dataphysics, Germany). When immersed in the measured liquid, the platinum plate was affected by the surface tension around it, which pulled the platinum plate down as far as possible. When the liquid surface tension and other related forces reached equilibrium with the balance force, the platinum sensing plate stopped its immersion into the liquid. Then, the balance sensor of the instrument measured the immersion depth and converted it into the surface tension value of the liquid. Each stabilizer solution was tested in triplicate.

The contact angle was measured by the sessile drop technique using a goniometer (OCA20, Dataphysics, Germany). Compressed disks of the compound powders were made under a 30 MPa compression force using a laboratory powder press (model 769YP-15A, Tian-jin, China). A droplet of purified water was placed onto the surface of each compressed disk and immediately observed through a low-power microscope until the drop reached a quasi-equilibrium shape. The contact angle was determined by measuring the tangent of the droplet on the disk surface.

The contact angle and surface tension (ST) values of the stabilizers such as hydroxypropylmethylcellulose (HPMC), polyvinylpyrrolidone K30 (PVP K30), sodium dodecyl sulfate (SDS), dioctyl sodium sulfosuccinate (DOSS), poloxamer 188 (P188), poloxamer 407 (P407), tween 80 (TW 80), tween 20 (TW 20), and carboxymethylcellulose sodium (CMC-Na) are given in Table 2 and we designed the experiment according to Table 3.

Based on the overall results of the trend analysis, it is now possible to develop a formulation design space with a critical set of API and stabilizer properties, from which generic formulations can be chosen for a stable nanosuspension [17]. This combination of properties can be used as a guide for choosing a stabilizer for the drugs because the most likely candidate for media milling was an API with high enthalpy and hydrophobicity, which can be stabilized either electrostatically or sterically [17]. The research showed that an active ingredient with low enthalpy (< 25 kJ/g) had a combination of properties that help guide the choice of stabilizer with a high viscosity and wetting index [18]. The higher the wetting index, the lower the ST and the smaller the contact angle. Therefore, we chose stabilizers with a low ST and a small contact angle. Products with a particle size of < 300 nm and a polydispersity index (PDI) of less than 0.25 were considered acceptable. After 1 h of milling, a significant reduction in the particle size was observed for the MDZ/NCs with 1% SDS, 1% DOSS, and 1% P407 (Table 3). The freshly milled nanosuspensions of the remaining formulations showed a narrow PSD, and the PDI was less than 0.25 for all the products (Table 4). All systems in this study gave a reproducible PSD, as observed in the second replicate studies. In this way, we developed a rational formulation design strategy in terms of the drug and stabilizer properties for the fabrication of stable MDZ/NCs.

### 2.2. Effect of Morphology: SEM Studies

Figure 1 shows that the MDZ coarse powders were minimized into nanoscale particles using the different stabilizers. Uniform and stable nanocrystals were obtained from the HPMC/SDS and HPMC/DOSS formulations.

### 2.3. Muscle Irritation Test

Muscle irritation and hemolysis tests were used to evaluate the potential toxicity issues of the MDZ/NCs (1% SDS) and MDZ/NCs (1% DOSS). The results showed that muscle fiber degeneration and inflammatory cell exudation were most pronounced after treatment with MDZ/NCs (1% DOSS). The greatest advantage of MDZ/NCs (1% SDS) is their low toxicity, which is related to their excipients. The results indicated that DOSS led to more severe muscle irritation than the equal-probability SDS. When MDZ and HPMC were added to both dosage forms, the severity of muscle irritation did not increase (Figure 2). Therefore, we chose the safer charge stabilizers, SDS and HPMC, to reduce muscle irritation.

### 2.4. Storage Stability

The particle sizes in the nanosuspensions of different drug-stabilizer combinations were plotted as a function of storage time (0–8 months, where 0 denotes freshly milled product) under three temperature conditions: 4, 25, and 40 °C, as shown in Figure 3. The original particle size was 286.6 ± 2.193 nm. The suspended particles remained relatively unchanged for up to 8 months of storage at all temperatures. The Ostwald ripening effect is a growing mechanism by which smaller particles dissolve and larger particles continue to grow, thereby increasing the average sizes of particles. The driving force of Ostwald ripening is the function of interface energy. Due to the dissolution of small particles and the growth of large particles, the specific interface energy per unit mass decreases, and the total free energy of the system decreases [19]. Neither Ostwald ripening nor simple aggregation occurred due to the protection of the energy barrier between the particles.

### 2.5. X-ray Powder Diffraction (XRPD)

The structural composition and crystalline state of the materials were determined by the XRD method. The milling process may cause changes in the physical form of the API. Attrition forces above certain values can cause increased lattice vibrations, resulting in crystal defects [15]. Peaks with a high intensity and narrow base widths are related to crystalline materials. The results from the XRD study showed that, following the milling process, the APIs remained crystalline with little to no amorphous content (Figure 4).

Figure 4. X-ray powder diffraction spectra illustrates the XRPD patterns of pure midazolam (API), MDZ/NCs (Nano), the physical mixture, and the blank excipient (Blank). Five diffraction peaks were observed for pure MDZ (API) between 2θ values of 10° and 40°, as shown by the green line. The MDZ/NCs prepared by wet milling showed diffraction peaks with lower intensity in the same range, representing a reduction between the crystalline form and the amorphous form. This indicates that both pure MDZ (API) and MDZ/NCs (Nano) existed in a crystalline state. The XRPD results confirmed that the crystalline state of the MDZ/NCs (Nano) was preserved and that there were no transformations during the media milling process.

### 2.6. DSC

The DSC determinations of pure midazolam (API) and MDZ/NCs (Nano) are shown in Figure 5. The API showed a characteristically sharp melting endothermic peak at 169 °C. The Nano sample also presented an endothermic peak at the same temperature. There was no significant difference between API and Nano; all of these data indicate that both had the same crystalline form.

### 2.7. Dissolution Behavior of MDZ/NCs

The dissolution release profiles for the MDZ/NCs, physical mixtures, and MDZ coarse powder were evaluated in PBS (pH 7.4). Figure 6 shows that the dissolution velocity of MDZ/NCs was distinctly superior compared to that of the physical mixture and the MDZ coarse powder. Within 8 h, 95.31% of the MDZ/NCs and 80.71% of the physical mixture were detected, while the amount of midazolam coarse powder was only 33.84% (Figure 6). In line with the Noyes–Whitney equation, the release rate was related to the relative surface area [20]. The smaller the particle size was, the higher the relative surface area was.

### 2.8. Pharmacokinetic Evaluation

The drug–time curve shows that, after the intramuscular injection of 1 mg/kg midazolam nanocrystals, there was an obvious absorption phase, and the plasma concentration of midazolam increased significantly at the initial stage of administration (Figure 7). The effects of both the MDZ/NCs and MDZ/S were rapid, and the peak time of the MDZ/NCs shows no obviously significant difference compared to that of the MDZ/S. As shown in Table 5, the clearance (CL) of the MDZ/NCs (563.02 ± 213.52 mL/h) was 1/3 lower than that of the MDZ/S (1545.07 ± 662.59 mL/h). The mean residence time (MRT) (0-t) of the MDZ/NCs (2.16 ± 0.74 h) was 3.13 longer than that of the MDZ/S (0.69 ± 0.19 h). In addition, The area of concentration-time curve (AUC) (0-t) of the MDZ/NCs (581.69 ± 225.05 ng/mL) was 2.72 times higher than that of the MDZ/S (217.01 ± 79.12 ng/mL). This may be because the nanoparticles increased the effect of drug adhesion on the biological mucosa and prolonged the retention time of drug adhesion, allowing the drug to be completely released in the absorption site to improve the characteristics of bioavailability.

### 2.9. In vivo Anticonvulsant Effect

Under the conditions of the selected doses (0.0625, 0.125, 0.25, 0.5, 1.0, and 2.0 mg), both MDZ/S and MDZ/NCs were able to significantly inhibit the occurrence of convulsions, and diazepam was able to significantly inhibit the occurrence of convulsions at high doses (1.0 mg, 2.0 mg). At medium and high doses (0.5, 1.0, and 2.0 mg/kg), midazolam nanocrystals inhibited the occurrence of convulsions, which reveals a better effect than that of the same dose of midazolam solution.

The anticonvulsant ED50 values and 95% confidence limits for the different drugs are given in Table 6. ED_50_ and 95% confidence limits for the anticonvulsant dose–effect curves for MDZ given by the different routes of administration at the two treatment times after MST convulsion onset. The fitted dose–effect curves determined from these results are displayed in Figure 8. When midazolam was given immediately after convulsion onset, the ED50 (0.017 mg/kg) for the MDZ/NCs was significantly more potent than that for MDZ/S (0.043 mg/kg), and both of the ED50 values were significantly more potent than the ED50 for DZP/S (0.212 mg/kg).

Convulsion termination latencies: All convulsion termination latencies for each drug, regardless of dose, were collapsed, and comparisons between the different drugs were evaluated by a Kruskal–Wallis one-way ANOVA based on the ranks for each treatment time. These differences were further evaluated using a Dunn’s multiple comparison test. Under the convulsion onset treatment condition, the convulsion termination latencies of the DZP/S-treated animals were significantly longer than those of the animals treated by MDZ/NCs or MDZ/S, while the latencies of MDZ/S and DZP/S were not significantly different (Figure 8). The EEG changes during convulsion and resting states are shown in Figure 9 below.

These data were collapsed across doses at each treatment time; the points represent data for individual animals, and the horizontal lines represent the medians. The MDZ/NCs produced significantly elevated (** p* < 0.05) latencies compared to both the MDZ/S (** p* < 0.05) and DZP/S (***** p* < 0.0001). MDZ effectively treated the convulsions caused by pentylenetetrazol (PTZ) and exerted an anticonvulsant effect. Figure 10 and Figure 11 show examples of this anticonvulsant effect. A Summary of the anticonvulsant effects of the MDZ treatment in the MST model is provided in Table 7.

### 2.10. Histopathological Analysis of Brain

The results showed that the neurons and glial cells in the hippocampus and the neurons in the prefrontal cortex and amygdala of normal rats were neatly arranged and that their structures were normal (Figure 12M–O).

In the MST model group, the neural cells exhibited a disordered arrangement, and many neural nuclei were pyknotic and necrotic, accompanied by an increase in small gelatinous fine cells and a large number of infiltrating inflammatory cells (Figure 12A–C). 

Some apoptosis was detected in both brain regions in the MDZ/NC-treated group (Figure 12D,E). In the visual field of the active control DZP/S group (Figure 12J–L), abundant pyramidal cells were observed in the hippocampus, and a large number of pyramidal cells were tightly arranged and deeply stained; the CA3 area exhibited the deepest staining, the boundary between the cytoplasm and the nucleus was unclear, as indicated by the black arrow, and some pyramidal cells were vacuolated, as indicated by the red arrows. No other obvious abnormalities were observed in the tissue.

The MDZ/NC-treated group indicated superior protective effects on brain tissues and high therapeutic efficacy. A morphological assessment of the brain sections revealed the formation of large edemic regions and cell apoptosis in the prefrontal cortex, the amygdala, and the hippocampus of MST rats that were either untreated (Figure 12A–C) or treated with DZP/S (Figure 12J–L). 

### 2.11. Oxidative Stress of Brain Tissue

According to Figure 13 and Table 8, MDZ/NCs significantly reduced the MDA content in the hippocampus of MST model rats and significantly increased the levels of glutathione and superoxide dismutase. This result suggests that the antiepileptic and brain-protective effects of midazolam may be related to its antioxidant capacity.

## 3. Discussion

Recent advances in milling technology have resulted in the reproducible production of drug particles in a size range of 200–300 nm [21,22]. Nanosuspension formulation possesses the distinct advantages of high drug loading, low excipient side effects, a low cost of production, and ease of scale-up [23]. Theoretically, a top-down nanosizing approach, such as wet milling, can be universally applied to most drug candidates [24,25].

The optimization of nanosuspensions using a media milling approach is a complex process involving many factors that affect the characteristics of the nanosuspension product [26]. Both the physical properties of the stabilizers and the API contribute to the stabilization mechanism and directly influence the formation of a product in the nanoscale range [27].

For APIs, log P and enthalpy are directly correlated with the feasibility of forming a stable nanosuspension. When selecting stabilizers, the type, compatibility, and dosage need to be carefully screened and evaluated [28]. It has been proposed that drugs with high molecular weights, low solubility, high melting points, and surface energies similar to those of stabilizers could be successfully processed into nanosuspensions [29].

The particle sizes and PDIs of the nanocrystals will affect the solubility, dissolution rate, and storage stability of the drug [30]. When particle size is controlled to within 150–300 nm, and the PDI is less than 0.25, the product is usually considered acceptable [31]. The zeta potential of the nanosuspension is also an important factor that affects stability. For nanosuspensions, the repulsion among nanoparticles also contributes to long-term stability [32]. It is generally believed that when electrostatic stabilization and steric stabilization are simultaneously present in a nanosuspension system, an absolute value of the zeta potential of 20 mV can effectively maintain the stability of the nanosuspension. In this study, based on the PSD data obtained immediately at the end of milling, the nanosuspension systems that successfully produced suitable particle sizes (along with their PDIs) were HPMC\DOSS, HPMC\SDS, and HPMC\P407 (Table 4). However, considering the zeta potential, only one formulation, HPMC\SDS, had an absolute zeta potential greater than 20 mV, which may yield a more stable nanosuspension system.

Nanosuspensions are thermodynamically unstable systems; hence, many problems affect the stability of nanosuspensions, including sedimentation, aggregation, crystal growth, and crystal transformation [33,34]. The sedimentation velocity of particles in the dispersion medium can be described by the Stokes equation (Equation (1) [35]):(1)$$V=2r^{2}\left(\rho_{1}-\rho_{2}\right)g/\left(9\eta \right)$$

According to the Stokes equation, reducing the sedimentation rate of the suspension can be achieved by reducing the particle radius, which can be achieved by preparing particles of a fine size and by increasing the viscosity of the dispersion medium to reduce the density difference between the solid particles and the dispersion medium, which requires the addition of suitable stabilizers.

The high relative surface area of the drug nanoparticles produces high surface energy, which results in aggregation. An electrostatic stabilizer is used to sufficiently wet the surface of the particles to form an electrostatic repulsion to create a high energy barrier as a result of inhibiting particle aggregation, as described by the colloidal stability (DLVO) theory [36]. According to the steric stabilization mechanism, the non-ionic surfactant adsorbs on the surface of the drug particles to cause steric hindrance among the particles, thereby promoting steric stability [37]. Moreover, the stabilizer’s adsorption on the surface of the nanoparticle without affecting the solubility of the drug will help reduce the interfacial tension between the solid particles and the liquid dispersion medium, thereby suppressing Ostwald ripening. This phenomenon can be described by the Ostwald–Freundlich equation (Equation (2) [36]):(2)$$log\left(S_{2}-S_{1}\right)=2\sigma M\left(1/r_{2}-1/r_{1}\right)/\rho RT$$

Due to differences in the preparation processes, drugs of the same chemical structure may exhibit different crystal forms, resulting in multiple types of one drug. Multi-type drugs may have different stabilities, dissolution properties, and bioavailability depending on the type, thereby affecting the safety of the drug. Therefore, it is necessary to study the crystal morphology of nanosuspensions; XRD and DSC are often used to confirm the crystalline form of a drug. XRD can distinguish between amorphous and crystalline forms. Crystalline drugs usually have a sharp melting point, which is not exhibited by amorphous particles. In addition, DSC can be used to distinguish among drug types and crystalline forms according to differences in their melting points. Therefore, DSC is often used as a supplement to XRD.

Improving the saturated solubility and dissolution rate of insoluble drugs is a major advantage of nanosuspensions. During the clinical use of nanosuspensions, the dissolution and release rate of the nanoparticles directly affects the rate and extent of drug absorption. Therefore, the dissolution and release rate, which can be determined in a specific dissolution medium by the dissolution method, are closely related to the efficacy of the drug [38]. The ability to formulate poorly-water-soluble molecules as nanometer-sized crystals can have a dramatic effect on bioavailability. Nanoparticles with higher surface areas can have a major impact on drug absorption. If bioavailability is truly limited by the dissolution rate, particle size reduction can significantly improve the performance of the drug [39]. The effects of particle size on the bioavailability of a poorly-water-soluble drug are demonstrated in Figure 7. In this study, the bioavailability of a poorly-water-soluble drug candidate was improved, as the particle size of the preparation was reduced to 200 nm using wet media milling technology. This pattern is routinely observed, provided that the primary factor affecting the bioavailability of the drug is the rate and extent of its dissolution. In conclusion, for poorly-water-soluble molecules, there are benefits to be gained when dissolution is no longer a limiting factor. The effects of both the MDZ/NCs and MDZ/S were rapid, and the peak time of the MDZ/NCs showed no obviously significant difference compared with that of the MDZ/S. The clearance (CL) of the MDZ/NCs (563.02 ± 213.52 mL/h) was 1/3 lower than that of the MDZ/S (1545.07 ± 662.59 mL/h). The MRT (0-t) of the MDZ/NCs (2.16 ± 0.74 h) was 3.13 longer than that of the MDZ/S (0.69 ± 0.19 h). In addition, The AUC (0-t) of the MDZ/NCs (581.69 ± 225.05 ng/mL) was 2.72 times higher than that of the MDZ/S (217.01 ± 79.12 ng/mL). In summary, the bioavailability of the nanocrystals of poorly-water-soluble compounds was significantly increased, and the drugs had a rapid effect and good absorption, which may be related to their pharmacokinetics and tissue distribution, which were characterized by a prolonged residence time in vivo after the intramuscular delivery of nanocrystals.

Epilepsy is a relatively common disease in neurology that is characterized by severe seizures that can cause damage to the brain; long-term seizures can even induce damage to the psychological state of patients, which not only affects their quality of life but also confers a large burden on their families and society [2,6]. There are many theories about the pathogenesis of epilepsy, among which central nervous system excitability and the dysfunction of neural inhibition are the most commonly accepted [37,40]. Studies have found that the dysfunction of neurons and glial cells, especially in the hippocampus, the frontal cortex, and the amygdala, can cause central nervous system excitability and the dysfunction of neural inhibition, thereby inducing epilepsy [41,42]. Glial cells can alter neurotransmitters, thus regulating ion channel switching, promoting the release of inflammatory factors, and effecting neuronal myelin and the microenvironment of neurons, thereby inducing epilepsy [43,44,45]. By observing the alignment and structural changes of neurons and glial cells in the hippocampus, the frontal cortex, and the amygdala, we can determine the protective effects of different drugs on the brain for combating seizures.

Histological observations have shown that neurons and glial cells in the hippocampus, the prefrontal cortex, and the amygdala of healthy rats exhibit an arranged order with a normal structure. In contrast, the neurons in the MST group were severely damaged; the pyramidal cell layer, cortical neurons, and the microglial nerve structure in the amygdala had essentially disappeared, and many cells became necrotic, forming red neurons. In the diazepam group, the pyramidal cell layer of the hippocampus and the cortical neurons and microglia in the amygdala were disordered, and nuclear pyknosis was significantly reduced. The microglia still increased, accompanied by the infiltration of neutrophils, although the outcomes were better than those of the MST group. The midazolam nanocrystalline group exhibited the best relative outcomes. The pyramidal cell layer in the hippocampus and the cortical neurons and microglia in the amygdala were arranged in an ordered manner; some of these microglia can be seen at high magnifications. The cells were swollen, denatured, and necrotic, there were more surviving cells, and their morphological structures were closer to normal. The midazolam nanocrystals exerted a protective effect on the hippocampus, the amygdala, and the prefrontal cortex, which resulted in better outcomes than treatments with diazepam and the midazolam solution.

The theory of oxidative stress injury was the initiating factor and is the most important link between epileptic attacks and brain injury. Oxidative stress caused by epilepsy first induces a large increase in oxygen radicals (ROS)/nitrogen radicals (RNS) [46]. Epilepsy directly damages the mitochondria and initiates the apoptosis process; additionally, epilepsy inhibits the antioxidant pathways, inhibits the expression of downstream antioxidant enzymes, and destroys the oxidation/antioxidant balance system in vivo, causing lipid peroxidation, oxidative stress aggravation, and neuronal damage.

MDA content increased significantly when lipid peroxidation occurred in vivo. This increase in MDA content reflected the severity of the injury level. MDA was degraded by a superoxide dismutase-mediated dismutation reaction. High SOD activity indicates a strong ability to scavenge oxygen free radicals [47]. Reductive GSH is an important reducing agent in vivo that directly acts on O_2_, H_2_O_2_, and LOOH and clears them to protect cells from free radical damage. The higher the GSH content is, the stronger the antioxidant capacity of the body [48]. Monika found that the MDA levels of rats with pentylenetetrazol acute convulsions were higher than those of the normal group. The GSH content of the amygdala was significantly reduced, and the MDA content was significantly increased [48]. The results of our study showed that the content of MDA in the hippocampus of epileptic rats increased significantly, while the GSH and SOD levels decreased significantly, indicating different degrees of oxidative stress injury in the brains of rats after an MST seizure.

## 4. Materials and Methods 

### 4.1. Materials

#### 4.1.1. Materials

MDZ was purchased from Yichang Renfu Pharmacy Company (Yichang, China). Diazepam was purchased from Shandong Xinyi Pharmaceutical Co., Ltd. (Dezhou, China). Poloxamer 407 (P407), poloxamer 188 (P188), Macrogol 4000 (PVP K30), and sodium dodecyl sulfate (SDS) were purchased from BASF, Ludwigshafen, Germany. Polysorbate 80 (TW80), polysorbate 20 (TW20), and carboxymethylcellulose sodium (CMC-Na) were ordered from Sinopharm Chemical Reagent Co. (Beijing, China). Sodium dodecyl sulfate was procured from VWR International (Radnor, PA, USA). Hydroxypropyl methylcellulose (HPMC) was purchased from Dow Chemical Company (Michigan, MI, USA). Dioctyl sodium sulfosuccinate (DOSS) was purchased from Hunan HuaNa Pharmaceutical Factory Chiral Pharmaceutical Co., Ltd. (Hunan, People’s Republic of China). HPLC-grade acetonitrile and methanol were obtained from Thermo Fisher Scientific (Waltham, MA, USA). All other reagents were of analytical grade. The purified water used in this study was prepared using a Mille-Q system (EMD Millipore, Billerica, MA, USA). The Cu/Zn-SOD assay kit with WST-8, the GSH assay kit, the lipid peroxidation MDA assay kit, and the BCA protein assay kit were purchased from Shanghai Biyuntian Biotechnology Co., Ltd. (Shanghai, China).

#### 4.1.2. Animals 

Male Sprague–Dawley rats weighing 180–220 g were provided by the Beijing Institute of Pharmacology and Toxicology (Beijing, China). Both the pharmacokinetic and pharmacodynamic studies were approved by the Animal Ethics Committee at Beijing Institute of Pharmacology and Toxicology (ethics code permit no. SCXK-(Beijing) 2007-004 and SCXK-(Beijing) 2007-003). Moreover, approval was received prior to beginning this research. The rats were acclimatized under standard conditions at a room temperature of 25 ± 2 °C and a relative humidity of 40%–60% under natural light/dark conditions for 1 week. The rats were provided food and water.

### 4.2. Preparation of MDZ/NCs

A wet milling technique was employed to prepare the nanosuspensions. First, we measured the viscosities and surface tensions of the different stabilizers. Then, we tested the contact angle between the midazolam and the stabilizers above. According to the properties of API and excipients, we knew how to design the experiments in theory. A high viscosity polymer material was used as the steric stabilizer and was combined with another steric stabilizer or electrostatic stabilizer. Each of the drug compounds was first dispersed in an aqueous HPMC (2.5%) solution containing one of the eight stabilizers: SDS, DOSS, TW80, TW20, P188, P407, PEG-4000, or CMC-Na (1.0%). The resulting suspension (5% drug content) was wet-milled with the milling media placed in a planetary mill (Dyno®-Mill Research Lab, WAB, Switzerland). The milling media used for this study consisted of 0.3 mm yttrium-stabilized zirconium beads. The optimum API concentration, stabilizer/API ratio, milling media size, milling time, and milling speed were identified in previous research and applied to this study. To obtain a uniform system, the mixed solution was pre-milled using Ultra Turrax (T25; IKA, Staufen, Germany) at 10,000 rpm for 10 min. The uniform suspensions were then milled at 1500, 2000, and 2500 rpm for 5 min, followed by 1 h at 3000 rpm to prepare the MDZ/NCs. The temperature of the preparation was 30 °C. Then, the average particle size (Z-average) and polydispersity index (PDI) were detected using a Malvern Zetasizer (ZS-90; Malvern Instruments, Malvern, UK).

### 4.3. SEM

The surfaces of hydrogels within or outside the MDZ/NCs were observed using an SEM (Hitachi SU8010; Hitachi Ltd., Tokyo, Japan). The hydrogel samples were freeze-dried and sputter-coated with a thin layer of gold under a vacuum. Microphotographs of the different samples were obtained with an electron beam under 15 kV energy. The magnification ranged from 10,000× to 15,000×.

### 4.4. Muscle Irritation Test

In brief, male rabbits weighing 2.5–3 kg were used and randomly divided into six groups (n = 3): MDZ/NCs (1% SDS), MDZ/NCs (1% DOSS), with an inactive ingredient of MDZ/NCs (2.5% HPMC and 1% SDS), an inactive ingredient of MDZ/NCs (2.5% HPMC and 1% DOSS), and 1.7% acetic acid solution (positive control). Each of the 5 groups was injected with 1 mL of the nanosuspension via quadriceps once daily for 5 days. The dose of MDZ was 1.0 mg/kg body weight. Paraffin-embedded tissues were cut, sectioned at 5 μm thickness, and stained with hematoxylin and eosin (H&E) for histopathological analysis.

### 4.5. Storage Stability

Three temperature conditions were applied in the stability study of the nanosuspensions: 4 °C (refrigerator), 25 °C, and 40 °C (thermostatic oven). Small aliquots of the suspension were periodically withdrawn to measure the particle size distribution (PSD) after 1–8 months of storage. The physical stability of the nanosuspensions was evaluated at 1–8 months of storage. The particle size measurements were performed using a Malvern Zetasizer.

### 4.6. X-ray Powder Diffraction (XRPD) Study

XRPD analysis was used to evaluate whether the initial crystalline state of the powders was maintained before and after particle size reduction. XRPD patterns of the samples were determined using a diffractometer (D8-Advance, Bruker, Germany) equipped with an Apex II CCD detector. The X-ray source was Kα radiation from a copper target with a graphite monochromator at a wavelength of 1.54 Å. Before this experiment, pure MDZ, MDZ/NCs, a blank excipient, and a physical mixture were analyzed. The sample was prepared by pressing the powder, which was fixed to a 0.3 mm cryoloop (Hampton Research, Aliso Viejo, CA). Because the sample was not suspended in any matrix, there was no background to interfere with the signal. The range (2θ) of scans was 5° to 50°, with a rate of 10°/min.

### 4.7. DSC

The DSC analysis of samples was performed using a Q2000 DSC apparatus (TA Instruments, New Castle, DE, USA). Five milligrams of pure MDZ, MDZ/NCs, a blank excipient, and a physical mixture were separately sealed in aluminum pans and scanned from 0 to 400 °C at a ramp rate of 10 °C/min under a nitrogen purge at a flow rate of 50 mL/min. An empty aluminum pan was used as the reference.

### 4.8. Drug Release Behavior

The dissolution behavior of the MDZ/NCs was evaluated by dialysis. Three groups, including the physical mixtures (MDZ, HPMC, and SDS), MDZ/NCs, and the MDZ powder, were diluted to 10 μg/mL by phosphate-buffered saline (PBS; pH 7.4). The drug was then released at 37 °C in 900 mL of PBS using a dissolution apparatus. At each time point, 2 mL of the sample was withdrawn from the dissolution chamber; a 0.22 μm filter was used to prevent any undissolved nanocrystals. The samples were characterized by high-performance liquid chromatography. The analysis was performed on a ZORBAX RX-C18 system (5 μm, 4.6 × 150 mm, Agilent, Santa Clara, USA) at 25 °C. The mobile phase was a methanol–phosphate buffer solution (dissolved 0.1 mol/L phosphate and 0.03 mol/L trimethylamine in 1000 mL water, adjusted to pH 3.5 with 1 mol/L sodium hydroxide) (65:35, *v*/*v*) at a flow rate of 1.0 mL/min. The detection wavelength was 220 nm, and the injection volume was 10 μL.

### 4.9. Pharmacokinetic Studies

According to the European Union’s guidelines for 18-year-old children, the dosage of midazolam (10 mg/person) for human use is approximately 0.15 mg/kg, and the dosage for rats is calculated to be 1.0 mg/kg according to the proportional difference between rat and human body weights.

Twelve male Sprague–Dawley rats were randomly and equally divided into two groups. Two different formulations, MDZ/NCs and the MDZ solution, were intramuscularly injected into the rats at a dose of 1.0 mg/kg. Orbital blood samples were collected using heparinized tubes at predetermined times and immediately centrifuged at 4000 rpm for 10 min. A total of 50 μL of plasma was separated from the supernatant by adding 50 μL methanol, 50 μL acetonitrile, and 50 μL of a propranolol acetonitrile solution (200 ng/mL). Then, the mixture was vortexed for 1 min and centrifuged at 14,000 rpm for 10 min. Finally, 5 μL of the supernatant was injected for LC/MS analysis to determine the MDZ concentration in the plasma.

### 4.10. In vivo Anticonvulsant Effect

Approximately 1 week before experimentation, the animals were implanted with stainless-steel cortical screw electrodes to record electroencephalographic (EEG) signals. The animals were anesthetized with pentobarbital and set in a stereotaxic frame. Three cortical stainless-steel screw electrodes were implanted in the animals’ skulls: two were placed bilaterally ~3.0 mm laterally from the midline and equidistant between bregma and lambda; the third was placed on the posterior calvarium as the reference electrode. Stainless-steel wires were used to attach the screws to a miniature connector plug. The electrodes, wires, and plugs were encased in cranioplastic cement. The incision was sutured; then, the animal was removed from the frame and placed on a warming pad for at least 30 min before being returned to the animal quarters.

The pentylenetetrazol (PTZ) seizure threshold model (MST) is a universal animal model of epilepsy. The number of convulsions in each group was observed. The behavioral responses to convulsions were classified by Racine’s classification: I: ipsilateral facial convulsions; II: nodding; III: contralateral forelimb clonus; IV: standing with clonic jumping; and V: falling with tonic–clonic seizures. When convulsions were considered to occur above IV or V, the number of convulsions in each group was recorded. The results showed that the number of convulsions in each group increased significantly by increasing the PTZ dose. According to the probit method, the dosage of PTZ was 69 mg/kg (ip) in 97% of the animals with paroxysmal convulsions. The rats were randomly divided into 18 groups, with 10 in each group. The MST model, MDZ/NCs, MDZ/S, and DZP/S were given at the same dose (0.0625, 0.125, 0.25, 0.5, 1.0, and 2.0 mg/kg), while the blank matrix was given at 0.2 mL/kg in the control group. The drugs were given immediately after convulsion onset via the intramuscular route. The number of rats with clonic convulsions was observed, and the latency of the convulsions was recorded to assess the anti-convulsive effect of the drugs on the MST model.

### 4.11. Histopathological Analysis of the Brain Sections

The Sprague Dawley rats were divided into the same five groups for the histopathological analysis: a negative control group, an MST model group, an MDZ/NCs group (1.0 mg/kg), an MDZ/S group (1.0 mg/kg), and a DZP/S group (1.0 mg/kg). The brains were fixed in glutaraldehyde (3%) for 48 h. After dehydration and embedding, the tissue was cut into 4 μm sections and stained with H&E. The sections were then visualized by light microscopy.

### 4.12. Oxidative Stress of the Brain Tissue

The SD rats were divided into the same five groups to test oxidative stress of the brain tissue (n = 6 each): negative controls (group 1), the untreated MST model (69 mg/kg PTZ; group 2); the MST model treated with MDZ/NCs (1.0 mg/kg; group 3); the MST model treated with MDZ/S (1.0 mg/kg; group 4); and the MST model treated with DZP/S (1.0 mg/kg; group 5).

The SD rats were decapitated twelve hours after the drugs were administered, and the hippocampus was quickly separated from the brain tissue and frozen at −80 °C. The frozen hippocampal tissue was added to saline in a 1:25 ratio, and the homogenate was prepared by mechanical homogenization under an ice water bath. The homogenate was then centrifuged for 10 min in a cryopreservation centrifuge for 4000 rpm. The concentration of the tissue protein was determined by a bicinchoninic acid (BCA) assay.

In an alkaline environment, the protein can be combined with Cu^2+^, reducing Cu^2+^ to Cu^1+^. BCA specifically combines with Cu^1+^ to form a stable purple blue complex, with a maximum absorption value at 562 nm. The color depth is directly proportional to the protein content. The protein concentration can be determined according to the absorption value. This specific operation was carried out according to the instructions provided in the BCA assay kit.

Thiobarbituric acid (TBA) can condense with malondialdehyde (MDA) to form a red product, which has a maximum absorption peak at 532 nm. We can calculate the MDA content by measuring the absorbance of the maximum absorption peak. This specific operation was carried out according to the instructions provided in the MDA assay kit.

Sulfhydryl compounds can react with dithiodinitrobenzoic acid to form a yellow compound with a maximum absorption peak at 420 nm. The content of GSH was calculated by measuring the absorbance of the maximum absorption peak. This specific operation was carried out according to the instructions provided in the GSH assay kit.

SOD has an inhibitory effect on superoxide anion free radicals, which can inhibit the oxidation of hydroxylamine by superoxide anion radicals and reduce the formation of nitrites, which are purplish red under the action of a chromogenic agent. The absorbance was determined by spectrophotometry, and the activity of the SOD in the sample was calculated by following the assay kit’s instructions.

### 4.13. Statistical Analysis

The dose–effect curves and the median effective dose (ED50) for the anticonvulsant activity of each individual drug were determined by probit analysis using four to seven doses with ten animals per group. A probit regression analysis (SPSS for Windows, Version 10.0, Chicago, IL, USA) was used to estimate the ED50s along with the 95% fiducial confidence limits for each drug treatment and agent combination. All quantitative results are presented as the means ± standard deviation. The statistical analysis was performed with Prism 6.0 (GraphPad Prism, San Diego, CA, USA). Comparisons between the control and treated groups were conducted using a one-way analysis of variance (ANOVA) with a Student’s *t*-test (two-tailed); *p* < 0.05 was considered statistically significant.

## 5. Conclusions

This study presents systematic research devised to address the drawbacks of insoluble drugs. We chose appropriate stabilizers according to the properties of MDZ and the wetting index between MDZ and different excipients in theory. The particle size of MDZ nanocrystals was maintained for 8 months, and our formulation has lower muscle irritation with 2.5% HPMC E5 and 1.0% SDS. Crystalline state analysis showed that the nanosizing process via wet milling had no influence on the crystalline state of MDZ. The dissolution rate increased significantly after dispersion compared to the physical mixture, and the bioavailability of MDZ was significantly improved by prolonging blood circulation. When MDZ was given immediately after convulsion onset, the ED_50_ (0.017 mg/kg) for MDZ/NCs was significantly more potent than that for MDZ/S (0.043 mg/kg). In histopathological analysis of the brain, MDZ/NCs significantly protected the brain from oxidative stress damage. In conclusion, nanocrystals dramatically enhanced the efficacy of MDZ in vitro and in vivo, and the particle size had a significant influence on the dissolution behavior, pharmacokinetic properties, and anticonvulsant and neuroprotective effects of nanocrystals. Therefore, the use of wet milling technology to formulate poorly water-soluble compounds is a viable approach capable of resolving many of the current issues associated with developing and commercializing poorly water-soluble molecules.

## Figures and Tables

**Figure 1 molecules-25-01080-f001:**
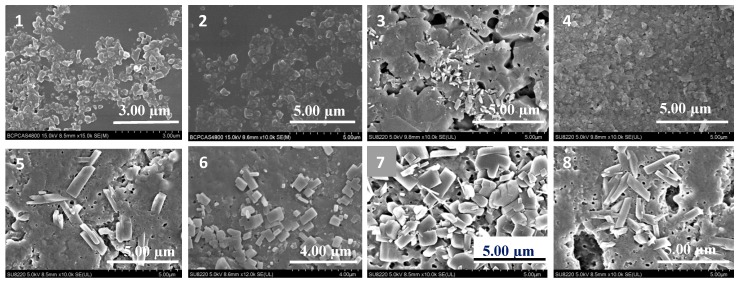
SEM images of the different formulations of MDZ/NCs. (**1**) HPMC/DOSS; (**2**) HPMC/SDS; (**3**) HPMC/PVP K30; (**4**) HPMC/CMC-Na; (**5**) HPMC/P407; (**6**) HPMC/P188; (**7**) HPMC/TW20; (**8**) HPMC/TW80.

**Figure 2 molecules-25-01080-f002:**
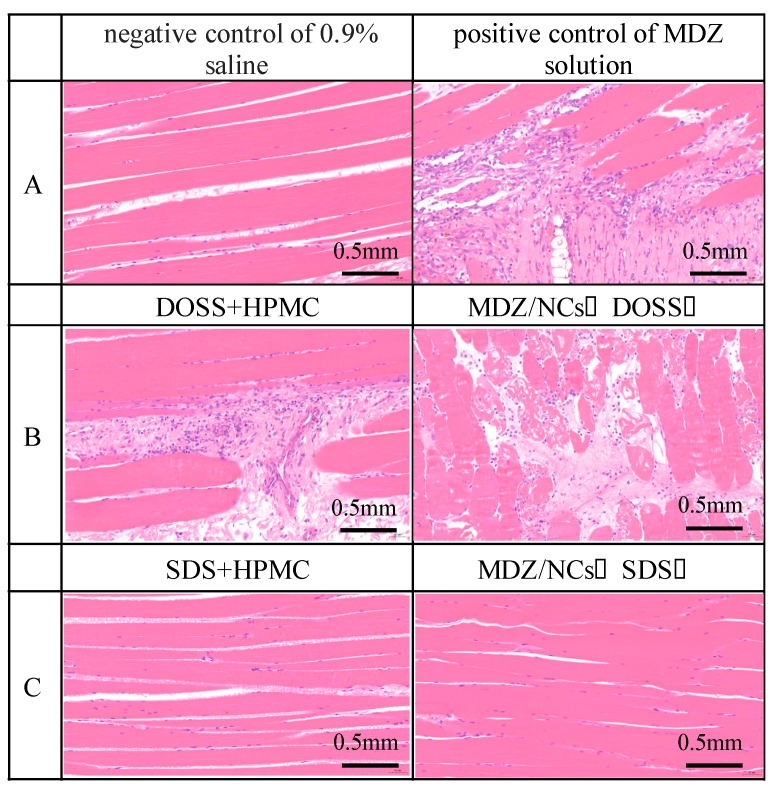
Comparison of muscle irritation with different formulations in rabbits. (**A**) control group: negative control with 0.9% saline; positive control with an MDZ solution with pH 2; (**B**) DOSS + HPMC; MDZ/NCs (DOSS); (**C**) SDS + HPMC; MDZ/NCs (SDS).

**Figure 3 molecules-25-01080-f003:**
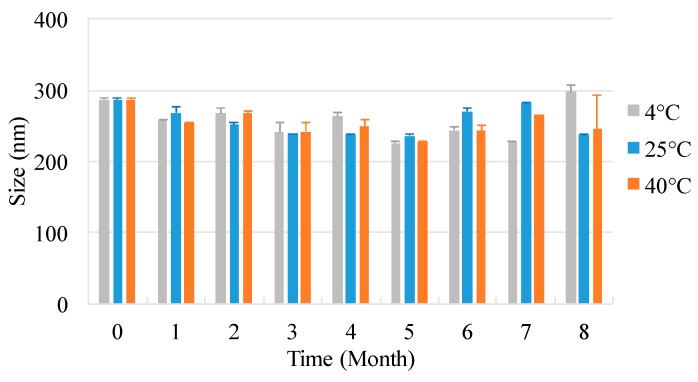
Stability of MDZ nanoparticles.

**Figure 4 molecules-25-01080-f004:**
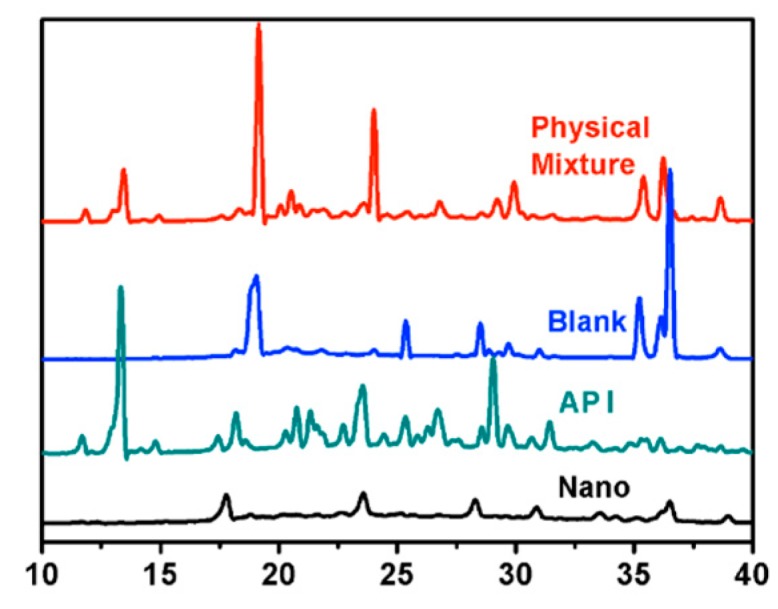
X-ray powder diffraction spectra. Abbreviations: Nano, MDZ nanocrystals; API, MDZ coarse powder.

**Figure 5 molecules-25-01080-f005:**
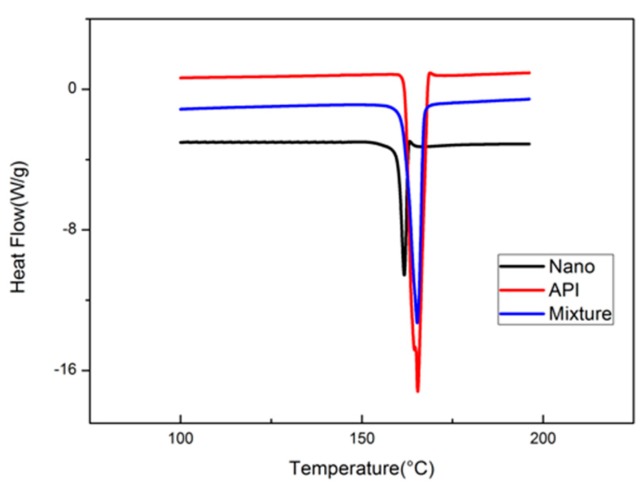
Reversible heat flow of the drug substances and nanosuspension variants using DSC. Abbreviations: Nano, MDZ nanocrystals; API, MDZ coarse powder; Mixture, physical mixture; Blank, blank excipients.

**Figure 6 molecules-25-01080-f006:**
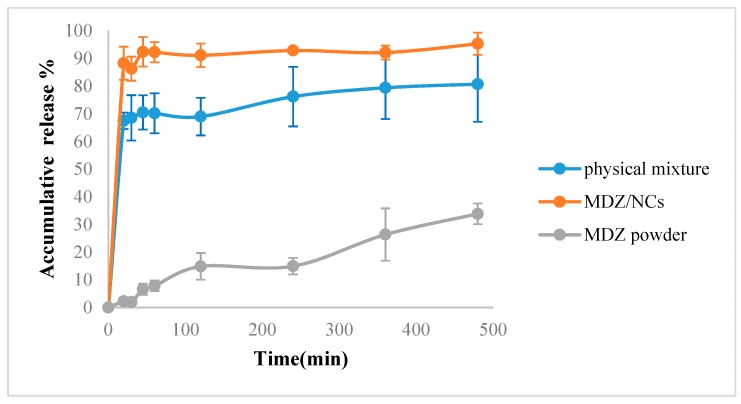
In vitro release profiles of the MDZ/NCs, the physical mixture, and the MDZ coarse powder in phosphate-buffered saline at pH 7.4. Abbreviations: MDZ/NCs, MDZ nanocrystals; MDZ/S, MDZ solution; DZP/S, diazepam solution.

**Figure 7 molecules-25-01080-f007:**
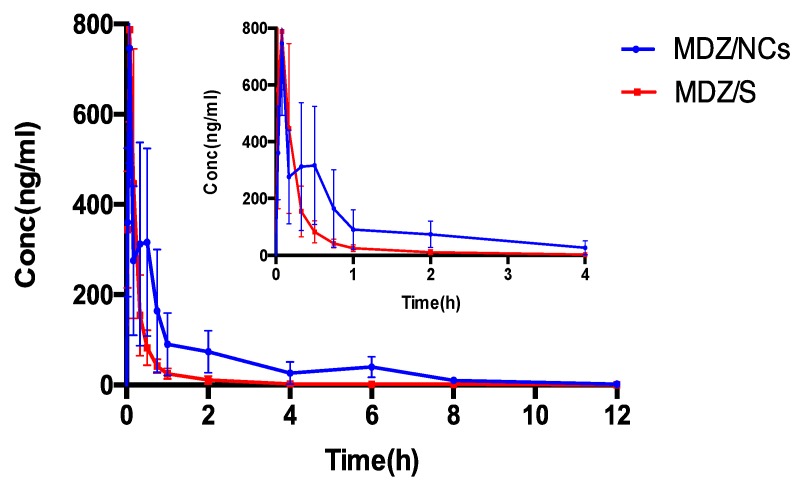
Plasma concentration–time curves for MDZ/NCs and MDZ/S, n = 6. Abbreviations: Nano, MDZ nanocrystals; Solution, MDZ solution.

**Figure 8 molecules-25-01080-f008:**
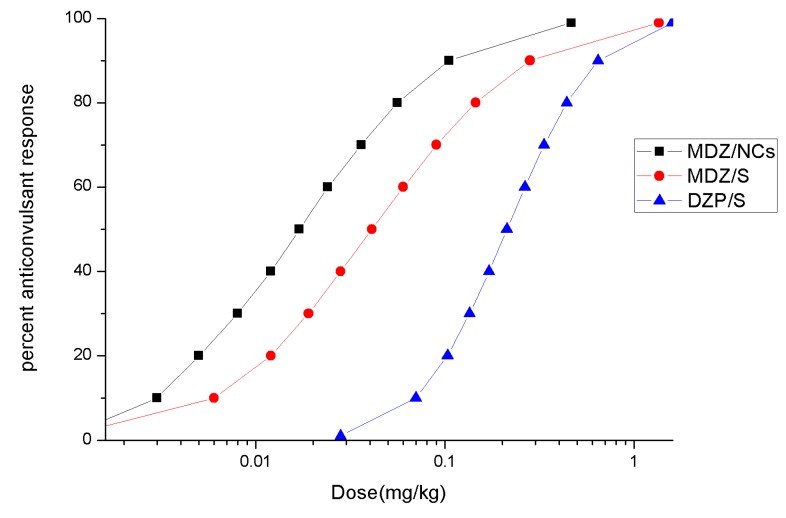
Fitted dose–effect curves calculated from the results of the probit analysis for anticonvulsant effects.

**Figure 9 molecules-25-01080-f009:**
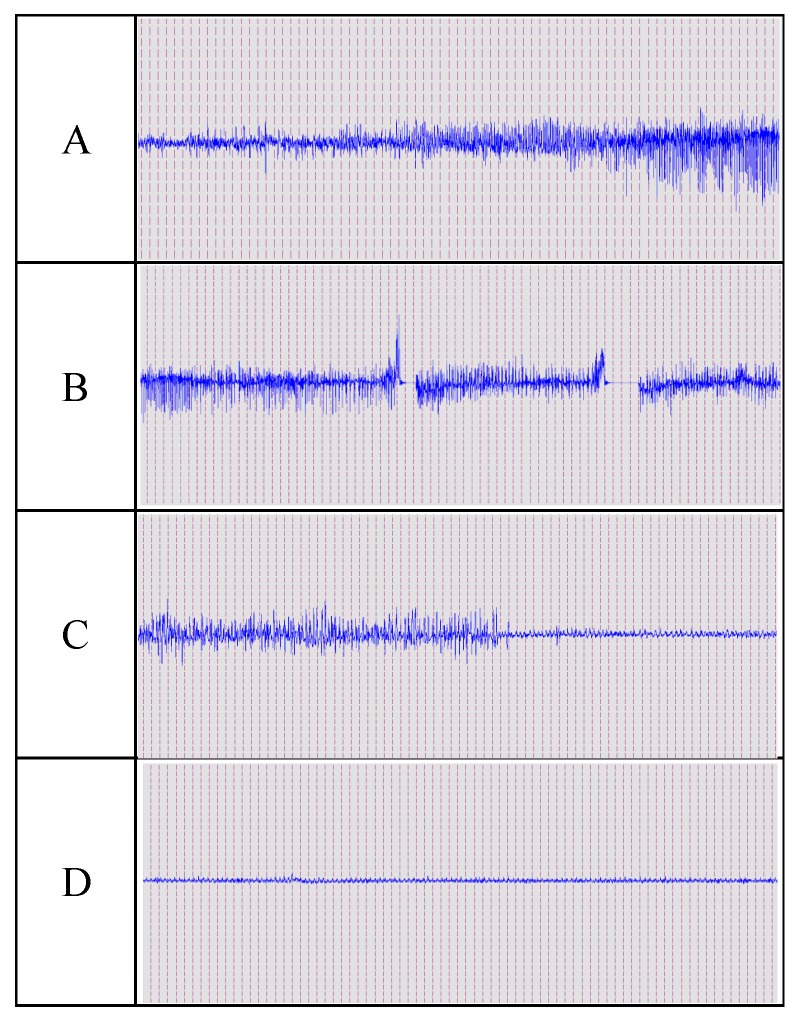
Example of midazolam control of PTZ-induced convulsion activity. The animal received an ED50 dose of MDZ/S or MDZ/NCs. (**A**) Gradually occurring convulsions. (**B**) Convulsions induced by PTZ. (**C**) The convulsion activity gradually stopped. (**D**) EEG after death of the animal. Calibration: 0.5 mV, 5 sec. Abbreviations: PTZ, pentylenetetrazole; EEG, electroencephalogram.

**Figure 10 molecules-25-01080-f010:**
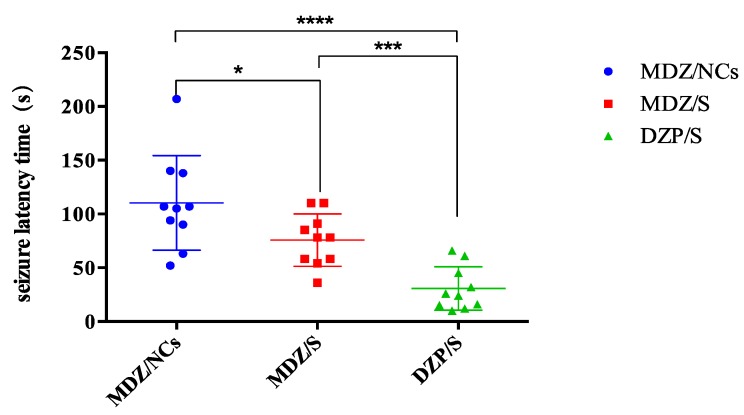
Convulsion latency time of the different formulations. Note: *****p* < 0.0001, MDZ/NCs vs. DZP/S; ****p* < 0.005, MDZ/S vs. DZP/S; **p* < 0.05, MDZ/NCs vs. MDZ/S.

**Figure 11 molecules-25-01080-f011:**
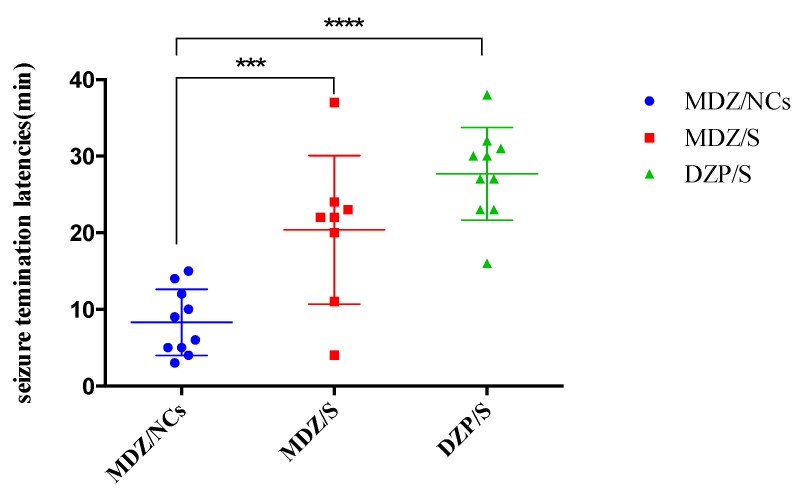
Convulsion termination latencies of the different formulations. Note: ****p* < 0.005, MDZ/NCs vs. MDZ/S; *****p* < 0.0001 MDZ/NCs vs. DZP/S; *p* > 0.05, MDZ/S vs. DZP/S.

**Figure 12 molecules-25-01080-f012:**
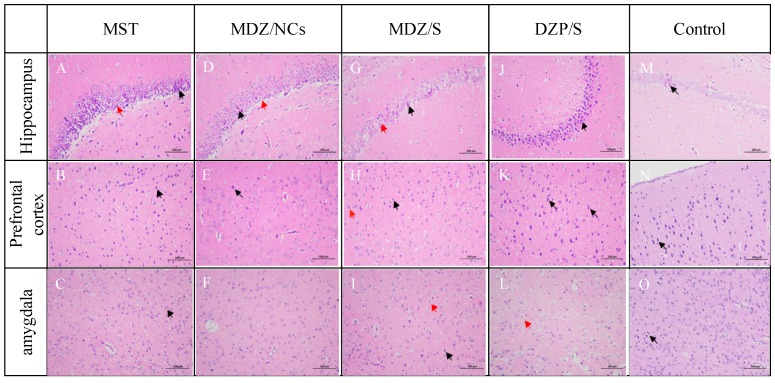
Histopathology of the hippocampus (**A**, **D**, **G**, **J**) and prefrontal cortex (**B**, **E**, **H**, **K**) of rats poisoned with pentrazol (**C**, **F**, **I**, **L**) and treated with MDZ/NCs (**D**–**F**), MDZ/S (**G**–**I**), and DZP/S (**J**–**L**); histopathology of the hippocampus of non-treated rats (MST) (**A**–**C**). The black arrows indicate pyknotic cells with strong cytoplasmic staining and nuclear condensation. The red arrows indicate edema. The images are shown at 200 × magnification.

**Figure 13 molecules-25-01080-f013:**
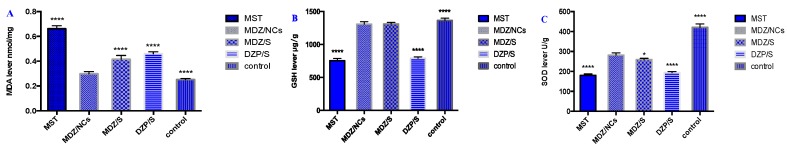
(**A**) Effect of MDZ/NCs on the MDA levels in the hippocampus of MST-induced convulsive rats. (**B**) GSH; (**C**) SOD. Note: The data represent the mean ± SD (n = 6); **p* < 0.05; *****p* < 0.01 compared with the MDZ/NC model group. Abbreviations: MDA, malondialdehyde; GSH, glutathione; SOD, superoxide dismutase.

**Table 1 molecules-25-01080-t001:** The attributes of midazolam.

	Properties						
	M.P./°C	Solubility (mg/mL)	pKa	log P	M.W.	Enthalpy (kJ/g)	Morphology
MDZ	159	0.024	5.5	4.33	325.771	15.7	White to light yellow crystalline

Abbreviation: M.P./°C, melting point; Solubility: the solubility in water; pKa, acidity coefficient; log P, oil water distribution coefficient; M.W., molecular weight.

**Table 2 molecules-25-01080-t002:** List of the stabilizers and their properties.

Drug	Properties					
	M.P./Tg *	CMC (%)	HLB	M.W.	ST (mN/m)	Contact Angle
HPMC	106	-	-	104100	32.16874	62.8
DOSS	153–157	0.29	10.2	444.56	24.50028	50.5
SDS	204–207	0.25	40	290–310	30.09747	26.8
CMC-Na	300	-	43.9	240	41.98717	63.2
PVPK30	130	-	14	3.8 *10^4^	32.82852	74.1
P188	52–57	0.017	29	102.13	32.60333	86.0
P407	56	0.02	18–23	12600	36.72650	86.0
T-80	−21	0.014	15	1310	34.50074	58.6
T-20	−21	0.031	16.7	1127.48	34.50074	34.9

Abbreviation: M.P./Tg *, melting point; CMC, critical micelle concentration; HLB, hydrophilic lipophilic equilibrium value; M.W., molecular weight; SDS, sodium dodecyl sulfate; DOSS, dioctyl sodium sulfosuccinate; HPMC, hydroxypropylmethylcellulose; P407, poloxamer 407; P188, poloxamer 188; TW20, tween 20; TW80, tween 80; PVP K30, polyvinylpyrrolidone K30; CMC-Na, carboxymethylcellulose sodium.

**Table 3 molecules-25-01080-t003:** Composition of the formulations.

	Drug (%)	Excipient								
		HPMC E5 (%)	DOSS (%)	SDS (%)	CMC-Na (%)	PVP K30 (%)	P188 (%)	P407 (%)	TW80 (%)	TW20 (%)
1	5	2.5	1							
2	5	2.5		1						
3	5	2.5			1					
4	5	2.5				1				
5	5	2.5					1			
6	5	2.5						1		
7	5	2.5							1	
8	5	2.5								1

**Table 4 molecules-25-01080-t004:** Results of the particle size and zeta potential.

#	Excipient	Results				
		Speed (rpm)	Time (h)	Size (nm)	PDI	Zeta Potential (mV)
1	HPMC\DOSS	3000	1	285.9	0.167	11.1
2	HPMC\SDS	3000	1	286.6	0.124	23.4
3	HPMC\CMC-Na	3000	1	370.8	0.315	55.5
4	HPMC\PVPK30	3000	1	381.2	0.21	4.34
5	HPMC\P188	3000	1	412	0.273	11.9
6	HPMC\P407	3000	1	275.9	0.187	13.8
7	HPMC\TW-80	3000	1	681.6	0.238	8.99
8	HPMC\TW-20	3000	1	315.8	0.208	12.6

**Table 5 molecules-25-01080-t005:** Pharmacokinetic parameters of MDZ/NCs-A and MDZ solution after intramuscular administration at a dose of 1.0 mg/kg in Sprague–Dawley rats (n = 6).

Parameters	Unit	MDZ/NCs	MDZ/S
T_1/2_	h	1.67 ± 0.46	1.48 ± 0.60
T_max_	h	0.15 ± 0.17	0.09 ± 0.04
C_max_	ng/mL	797.20 ± 236.88	805.62 ± 204.71
AUC(0-t)	h·ng/mL	581.69 ± 225.05 **	217.01 ± 79.12
AUC(0-∞)	h·ng/mL	591.46 ± 219.16 **	219.82 ± 78.59
V	mL	1392.24 ± 656.24	3279.63 ± 1708.20
CL	mL/h	563.02 ± 213.52 **	1545.07 ± 662.59
MRT(0-t)	h	2.16 ± 0.74 **	0.69 ± 0.19
MRT(0-∞)	h	2.50 ± 0.72 **	0.89 ± 0.34

Note: The data represent the mean ± SD (n = 6) (** *p* < 0.01) compared with the same treatment dose in the MDZ/NC model group. Abbreviations: CL, clearance; MRT, mean residence time; AUC_(0–t)_, area under the curve from time of administration to time “t”; AUC_(0–∞)_, area under the curve from time of administration through infinite time; V, apparent distribution volume; C_max_, peak plasma concentration.

**Table 6 molecules-25-01080-t006:** ED_50_ and 95% confidence limits for the anticonvulsant dose–effect curves for MDZ given by the different routes of administration at the two treatment times after MST convulsion onset.

Route	Drug	ED_50_ (mg/kg)	95% Confidence Limits (mg/kg)
Intramuscular	MDZ/NCs	0.017 mg/kg	0.001–0.043
	MDZ/S	0.043 mg/kg	0.001–0.098
	DZP/S	0.212 mg/kg	0.139–0.341

Abbreviations: MDZ/NCs, MDZ nanocrystals; MDZ/S, MDZ solution; DZP/S, diazepam.

**Table 7 molecules-25-01080-t007:** Summary of the anticonvulsant effects of the MDZ treatment in the MST model.

Group	Convulsion Latency Time (s)	Convulsion Termination Latencies (min)
MDZ/NCs	110.30 ± 13.89	8.30 ± 1.37
MDZ/S	75.80 ± 7.72	20.38 ± 3.43
DZP/S	30.70 ± 6.40	27.70 ± 1.92

**Table 8 molecules-25-01080-t008:** Protective effects of MDZ/NCs on the hippocampus.

Group	Concentration		
	MDA (nmol/mg)	GSH (μg/g)	SOD (U/g)
MST	0.6615 ± 0.0244 ****	1311.394 ± 25.100 ****	181.48 ± 4.875 ****
MDZ/NCs	0.4156 ± 0.1493	791.671 ± 21.152	276.276 ± 8.630
MDZ/S	0.4603 ± 0.1544 ****	753.148 ± 36.238	268.810 ± 6.432 *
DZP/S	0.2993 ± 0.1641 ****	1308.508 ± 37.736 ****	193.924 ± 5.183 ****
Control	0.2527 ± 0.0778 ****	1367.027 ± 33.084 ****	420.014 ± 17.790 ****

Note: *****p* < 0.0001, **p* < 0.05, vs. MDZ/NCs.

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
