# Peer review of "Preparation of Nanocrystals for Insoluble Drugs by Top-Down Nanotechnology with Improved Solubility and Bioavailability"

_molecules, 2020, doi:10.3390/molecules25051080_

Round 1

Reviewer 1 Report

As far as the manuscript concerned, I think it is a good and complete work because they have performed different studies to proof what they claim (formulation, toxicity, bioavailability, pharmacokinetics, etc) and it could be Pharmaceutical Industry interest of. But there are some considerations and questions the authors have to have into account for considering to publish:

  • The methods to study the stabilizer properties together with the active are not well describe in the paper, in the section of methods
  • In table 1 the authors describe the results of active characteristics, but I don’t know if they have found them experimentally or by literature, this is not clear. Also, although, I understand that solubility has been carried out in water, but it is not indicated.
  • There are some messages along the results “Error! Reference source not found” that at the end I’ve found out they are figures or tables, but I’m not sure If I’ve correlated well each messages:Page 2 line 78: Table 3Page 4, line 112: Figure 1Page 5, line 140: Figure 3Page 7, line 182: figure 6Page 10, line254: Figure 10 and figure 11Page 11, line 270: figure 11Page 11, line 278: figure 12Page 11, line 280: figure 12Page 12, line 290: figure 12
  • Page 12, line 299: figure 13 and table 8
  • Page 12, line 289: figure 12
  • Page 11, line 279: figure 12
  • Page 11, line 275: figure 12
  • Page 11, line 269: Figure 10
  • Page 7, line 185: figure 6
  • Page 7, line 170: figure 5
  • Page 4, line 129: Figure 2
  • Page 3, line 101: Table 4
  • Page 2 line 78: table 2
  • Page 5, lines 141 and 142: The information that appear in these lines is the same as the information under figure 3 (as a footnote), please eliminate because I think it is an error.
  • Figure 9 are not mentioned in the text, so if it is not relevant, please don’t include it.
  • Page 11, line 264: It is indicated that is table 1, but I think it is table 7. It is a writing mistake.
  • In page 15, in the method part the authors talk about Sodium dodecyl sulfate and Lauryl sodium sulfate as two different entities, but they are the same. So, indicate it only once and put together where they have been purchased.
  • In 4.4 part, lines 484 and 485, I think the information is duplicate, is it possible that one of them were inactive ingredient of MDZ/NCs (2.5 % HPMC and 1 % DOSS)?
  • Page 18, line 550: Where convulsions were considered to occur above IV or IV? Above IV or V?
  •  
  • In conclusion, taking in account, all the consideration explained before, in my opinion, article could be consider for publishing at your journal.

Author Response

Point 1: The methods to study the stabilizer properties together with the active are not well describe in the paper, in the section of methods.

Response 1:

I added the specific preparation in the method: First, we measured the viscosities and surface tensions of the different stabilizers. Then, we tested the contact angle between the midazolam and the stabilizers above. According to the properties of API and excipients, we knew how to design the experiments in theory. A high viscosity polymer material was used as the steric stabilizer and was combined with another steric stabilizer or electrostatic stabilizer. Each of the drug compounds was first dispersed in an aqueous HPMC (2.5%) solution containing one of the eight stabilizers: SDS, DOSS, TW80, TW20, P188, P407, PEG-4000, or CMC-Na (1.0%).

Point 2: In table 1 the authors describe the results of active characteristics, but I don’t know if they have found them experimentally or by literature, this is not clear. Also, although, I understand that solubility has been carried out in water, but it is not indicated.

Response 2:

  • I have added the note under the table 1: Solubility: the solubility in water.
  • I found the properties of midazolam (M.P., solubility, pKa, log P, M.W., morphology) in the pubchem database. The URL is https://pubchem.ncbi.nlm.nih.gov/compound/4192#section=WHO-ATC-Classification-System
  • The enthalpy is calculated by the formula[1]: Hm25°C= 23.7Tb + 0.020Tb2−2950

       Hrepresents molar evaporation energy of compound,Trepresents the boiling point of the compound.

[1] Peng-Fei Yuea,b,∗, Yu Lia, Jing Wana, Ming Yanga,∗, Wei-Feng Zhua, Chang-Hong Wangb, Study on formability of solid nanosuspensions during nanodispersion and solidification: I. Novel role of stabilizer/drug property, International Journal of Pharmaceutics 454 (2013) 269–277.

Point 3: Figure 9 are not mentioned in the text, so if it is not relevant, please don’t include it.

Response 3: I have added that: The EEG changes during convulsion and resting state are shown in the figure 9 below.

Point 4: In page 15, in the method part the authors talk about Sodium dodecyl sulfate and Lauryl sodium sulfate as two different entities, but they are the same. So, indicate it only once and put together where they have been purchased.

Response 4: I have modified “lauryl sodium sulfate” to sodium dodecyl sulfate.

Point 5: In 4.4 part, lines 484 and 485, I think the information is duplicate, is it possible that one of them were inactive ingredient of MDZ/NCs (2.5 % HPMC and 1 % DOSS)?

Response 5: I have modified “MDZ/NCs (2.5% HPMC and 1% SDS)” to MDZ/NCs (2.5% HPMC and 1 % DOSS).

Point 6: Page 18, line 550: Where convulsions were considered to occur above IV or IV? Above IV or V?

Response 6: I have modified the “IV or IV” to IV or V.

Point 7: There are some messages along the results “Error! Reference source not found” that at the end I’ve found out they are figures or tables, but I’m not sure If I’ve correlated well each messages: Page 2 line 78: Table 3Page 4, line 112: Figure 1Page 5, line 140: Figure 3Page 7, line 182: figure 6Page 10, line254: Figure 10 and figure 11Page 11, line 270: figure 11Page 11, line 278: figure 12Page 11, line 280: figure 12Page 12, line 290: figure 12; Page 12, line 299: figure 13 and table 8; Page 12, line 289: figure 12; Page 11, line 279: figure 12; Page 11, line 275: figure 12; Page 11, line 269: Figure 10; Page 7, line 185: figure 6; Page 7, line 170: figure 5; Page 4, line 129: Figure 2; Page 3, line 101: Table 4; Page 2 line 78: table 2

Response 7: I am sorry for the mistakes. Maybe it resulted form the format confusion after upload. Thanks for your understanding.

Point 8: Page 5, lines 141 and 142: The information that appear in these lines is the same as the information under figure 3 (as a footnote), please eliminate because I think it is an error.

Response 8: It is an error and I have deleted the same information.

Point 9: Page 11, line 264: It is indicated that is table 1, but I think it is table 7. It is a writing mistake.

Response 9: I have modified “table 1” to “table 7”.

Reviewer 2 Report

This manuscript describes the preparation and properties of nanocrystals of the benzodiazepine drug Midazolam. The latter is known to inhibit convulsion and epilepsy, but its practical application has been limited by its low solubility under physiological conditions. Similar bio-uptake restrictions also apply to very many other potentially beneficial drug compounds.

The Midazolam nanocrystals are readily prepared, form a stable nanosuspension that causes little muscle irritation, and exhibit significantly improved bioavailability. This present paper is important in overcoming the earlier problems, but has considerably wider significance with respect to poorly soluble pharmaceuticals in general. The work has been carried out extremely thoroughly and is described very clearly. The paper should be accepted for publication after the revisions noted below have been dealt with.

In writing a scientific manuscript it is customary practice to ensure that all the individual Tables and Figures are mentioned in the main text. This has generally been done in this manuscript, though there may be a few omissions. The situation has been complicated by apparent application of an editorial program that has changed many of these text citations into Error! inserts. This is no fault of the authors. It also makes the referees’ job very difficult in this context. The authors should check carefully that their final manuscript contains all the necessary Table and Figure citations in the main text.

The manuscript describes a major investigation. Very sensibly, much use is made of abbreviations, like SDS or TW20, for brevity. Most of these abbreviations are clearly defined. On the other hand, these abbreviation explanations are sometimes repeated several times in different Table captions. One use is generally sufficient. The TW80 abbreviation should read: TW80, tween 80.

I presume that ST is surface tension. However, its units of measurement (see Table 1) appear to be incorrectly given. I think this should be millinewtons per metre (mN/m).

Nanoscale particles have a tiny surface area, but have high surface area relative to the number of their constituent molecules. I think the correct term is high relative surface area. See, for example, Page 2 lines 45 & 46; and Page 14, line 351.

Minor alterations:

Page 1, Title: Preparation of nanocrystals for ...

Page 1, line 35: drug resistance [4].

Page 1, line 40: intramolecular application of midazolam

Page 11, Table heading, line 264: Table 7 (not Table 1)

Page 15, lines 401 & 402: The word alter is used three times in the same sentence here. Re-write this sentence.

Page 15, line 448: Docusate may be a local trade name? Probably better to call it Dodecyl sulfate.

Page 16, line 467: Delete the red numerals 21?

Page 19, Ref. 1, line 634: Full publication details required

Page 20, Ref. 17, line 669: Year of publication?

Page 20, Ref. 18, line 671: Journal name? Year?

Author Response

Point 1: In writing a scientific manuscript it is customary practice to ensure that all the individual Tables and Figures are mentioned in the main text. This has generally been done in this manuscript, though there may be a few omissions. The situation has been complicated by apparent application of an editorial program that has changed many of these text citations into Error!inserts. This is no fault of the authors. It also makes the referees’ job very difficult in this context. The authors should check carefully that their final manuscript contains all the necessary Table and Figure citations in the main text.

Response 1: I am very sorry for that. It may because the format confusion after upload. I have modified the mistakes. Thank you for your understanding.

Point 2: The manuscript describes a major investigation. Very sensibly, much use is made of abbreviations, like SDS or TW20, for brevity. Most of these abbreviations are clearly defined. On the other hand, these abbreviation explanations are sometimes repeated several times in different Table captions. One use is generally sufficient. The TW80 abbreviation should read: TW80, tween 80.

Response 2: I have deleted the redundant abbreviations of Table 3, Table 4, Table 8, Figure 1, Figure 8, Figure 10, Figure 11, Figure 12. I have modified the mistake “TW80, tween 20” to “TW80, tween 80” under Table 2.

Point 3: I presume that ST is surface tension. However, its units of measurement (see Table 1) appear to be incorrectly given. I think this should be millinewtons per metre (mN/m).

Response 3: I have modified the mistake “Mm/m” to “mN/m”.

Point 4: Nanoscale particles have a tiny surface area, but have high surface area relative to the number of their constituent molecules. I think the correct term is high relative surface area. See, for example, Page 2 lines 45 & 46; and Page 14, line 351.

Response 4: I have modified the “enormous” and “huge” to “high relative”.

Point 5: Page 1, Title: Preparation of nanocrystals for ...

Response 5: I have modified title “Preparation nanocrystals” to “Preparation of nanocrystals”.

Point 6: Page 1, line 35: drug resistance [4].

Response 6: I have modified “drug resistence” to “drug resistance”.

Point 7: Page 1, line 40: intramolecular application of midazolam

Response 7: I have modified “intramuscular” to “intramolecular application of midazolam”.

Point 8: Page 11, Table heading, line 264: Table 7 (not Table 1)

Response 8: I have modified “Table 1” to “Table 7”.

Point 9: Page 15, lines 401 & 402: The word alter is used three times in the same sentence here. Re-write this sentence.

Response 9: I have re-write the sentence: “Glial cells can alter neurotransmitters, thus regulating ion channel switching, promoting the release of inflammatory factors, and effecting neuronal myelin and the microenvironment of neurons, thereby inducing epilepsy.”

Point 10: Page 15, line 448: Docusate may be a local trade name? Probably better to call it Dodecyl sulfate.

Response 10: I have modified the “docusate” to “Dioctyl sodium sulfosuccinate”.

Point 11: Page 16, line 467: Delete the red numerals 21?

Response 11: I have deleted number 21.

Point 12: Page 19, Ref. 1, line 634: Full publication details required

Response 12:

Ref. 1 Thijs, R. D.; Surges, R.; O'Brien, T. J.; Sander, J. W., Epilepsy in adults. Lancet 2019, 393, (10172), 1-13.

Point 13: Page 20, Ref. 17, line 669: Year of publication?

Response 13:

Ref. 17 Owen, H.; Graham, S.; Werling, J. O.; Carter, P. W., Anion effects on electrostatic charging of sterically stabilized, water insoluble drug particles. Int J Pharm 2009, 368, (1-2), 154-159.

Point 14: Page 20, Ref. 18, line 671: Journal name? Year?

Response 14:

Ref. 20(Ref. 18 before modified)Mauludin, R.; Müller, R. H.; Keck, C. M., Kinetic solubility and dissolution velocity of rutin nanocrystals. Eur J Pharm Biopharm 2009, 36, (4-5), 502-510.

Reviewer 3 Report

Paper is really interesting and deals with the studies on the preparation of formulations for poorly water-soluble drugs such as e.g. midazolam. Authors firstly provide an adequate background of the research and then present extensive investigations on the development of the most effective systems followed by the deep analysis of obtained results. Manuscript is worth considering for publication but it requires some minor revisions which are listed below.

Unexplained abbreviations such as GSH or SOD should be removed from abstract and replaced by the whole names. Some methods applied by Authors have been only mentioned and need to be briefly described, e.g. Wilhelmy plate method used for the determining the surface tension of the stabilizer solutions or BCA assay. Additionally, the phenomenon of Ostwald ripening mentioned in 2.4. section needs to be characterized in few sentences. In section 2.1. it was stated by Authors that the stabilizers with low ST and a small contact angle have been chosen. Brief explanation telling why exactly such parameters were decisive should be added. Section 4.2. should be supplemented with information concerning the temperature in which the midazolam nanocrystals were prepared. Two tables presented in the paper are marked as Table 1. From the editorial point of view, there should be a space between a text and a citation placed in square brackets. Section Conclusions is too extensive; it should briefly present the main highlights of the research. Section References: as recommended by the requirements of the Journal, all references should contain abbreviations of the journal titles instead of their whole names.

Author Response

Point 1: Unexplained abbreviations such as GSH or SOD should be removed from abstract and replaced by the whole names. 

Response 1: I have modified the abbreviations “GSH”, “SOD”, “MDA” to “glutathione”, “superoxide dismutase” and “malondialdehyde”.

Point 2: Some methods applied by Authors have been only mentioned and need to be briefly described, e.g. Wilhelmy plate method used for the determining the surface tension of the stabilizer solutions or BCA assay. 

Response 2: I have supplemented the experimental method

Wilhelmy plate method: Surface tension measurements of the stabilizer solutions were determined using the Wilhelmy plate method on a DCAT21 instrument (Dataphysics, Germany). When immersed in the measured liquid, the platinum plate was affected by the surface tension around it, which pulled the platinum plate down as far as possible. When the liquid surface tension and other related forces reached equilibrium with the balance force, the platinum sensing plate stopped its immersion into the liquid. Then, the balance sensor of the instrument measured the immersion depth and converted it into the surface tension value of the liquid. Each stabilizer solution was tested in triplicate.

BCA assay: In alkaline environment, the protein can be combined with Cu2+, reducing Cu2+ to Cu1+. BCA specifically combines with Cu1+ to form a stable purple blue complex, with a maximum absorption value at 562 nm. The color depth is directly proportional to the protein content. The protein concentration can be determined according to the absorption value. This specific operation was carried out according to the instructions provided in the BCA assay kit.

Point 3:  Additionally, the phenomenon of Ostwald ripening mentioned in 2.4. section needs to be characterized in few sentences.

Response 3:

I have explained the theory of Ostwald ripening : The Ostwald ripening effect is a growing mechanism by which smaller particles dissolve and larger particles continue to grow, thereby increasing the average sizes of particles. The driving force of Ostwald ripening is the function of interface energy. Due to the dissolution of small particles and the growth of large particles, the specific interface energy per unit mass decreases, and the total free energy of the system decreases.

Point 4: In section 2.1. it was stated by Authors that the stabilizers with low ST and a small contact angle have been chosen. Brief explanation telling why exactly such parameters were decisive should be added.

Response 4: I have added the explanation:

The research of Peng-Fei Yue et al. showed that an active ingredient with low enthalpy (<25 kJ/g) had a combination of properties that help guide the choice of stabilizer with a high viscosity and wetting index. [18] The higher the wetting index, the lower the ST and the smaller the contact angle. Therefore, we chose stabilizers with a low ST and a small contact angle.

Point 5: Section 4.2. should be supplemented with information concerning the temperature in which the midazolam nanocrystals were prepared. 

Response 5:

I have added that: “The temperature of the preparation was 30 °C.” Because the temperature is related to the milling speed. The milling speed of 3000rpm is almost stationary in 30 °C.

Point 6: Two tables presented in the paper are marked as Table 1.

Response 6: I've fixed the mistake.

Point 7: From the editorial point of view, there should be a space between a text and a citation placed in square brackets. 

Response 7: I've added a space between the citation placed in square brackets and the text.

Point 8: Section Conclusions is too extensive; it should briefly present the main highlights of the research. 

Response 8: I have modified the conclusion briefly.

This study presents systematic research devised to address the drawbacks of insoluble drugs. We chose appropriate stabilizers according to the properties of MDZ and the wetting index between MDZ and different excipients in theory. The particle size of MDZ nanocrystals was maintained for 8 months, and our formulation has lower muscle irritation with 2.5% HPMC E5 and 1.0% SDS. Crystalline state analysis showed that the nanosizing process via wet milling had no influence on the crystalline state of MDZ. The dissolution rate increased significantly after dispersion compared to the physical mixture, and the bioavailability of MDZ was significantly improved by prolonging blood circulation. When MDZ was given immediately after convulsion onset, the ED50 (0.017 mg/kg) for MDZ/NCs was significantly more potent than that for MDZ/S (0.043 mg/kg). In histopathological analysis of the brain, MDZ/NCs significantly protected the brain from oxidative stress damage. In conclusion, nanocrystals dramatically enhanced the efficacy of MDZ in vitro and in vivo, and the particle size had a significant influence on the dissolution behavior, pharmacokinetic properties, and anticonvulsant and neuroprotective effects of nanocrystals. Therefore, the use of wet milling technology to formulate poorly water-soluble compounds is a viable approach capable of resolving many of the current issues associated with developing and commercializing poorly water-soluble molecules.

Point 9: Section References: as recommended by the requirements of the Journal, all references should contain abbreviations of the journal titles instead of their whole names.

Response 9: I have fixed the mistakes.

This manuscript is a resubmission of an earlier submission. The following is a list of the peer review reports and author responses from that submission.